# Comprehensive Analysis and Greenhouse Gas Reduction Assessment of the First Large-Scale Biogas Generation Plant in West Africa

Haoran Chen [1], Qian Xu [1], Shikun Cheng [1,*], Ting Wu [2], Tong Boitin [2], Sunil Prasad Lohani [3], Heinz-Peter Mang [4], Zifu Li [1] and Xuemei Wang [1,*]

1   Beijing Key Laboratory of Resource-Oriented Treatment of Industrial Pollutants, School of Energy and Environmental Engineering, University of Science and Technology Beijing, Beijing 100083, China; m202120173@xs.ustb.edu.cn (H.C.); 41926036@xs.ustb.edu.cn (Q.X.); zifuli@ustb.edu.cn (Z.L.)
2   Chengdu DeTong Environmental Engineering Co., Ltd., Shenglong Str. 9, Offices 301–305, Chengdu 610041, China; timmy.wu@chengdu-detong.com (T.W.); t.boitin@chengdu-detong.com (T.B.)
3   Renewable and Sustainable Energy Laboratory, Department of Mechanical Engineering, Kathmandu University, Dhulikhel 45200, Nepal; splohani@ku.edu.np
4   German Society for Sustainable Biogas and Bioenergy Utilization (GERBIO), Kirchberg an der Jagst, 74592 Baden, Germany; hpmang@t-online.de
*   Correspondence: chengshikun@ustb.edu.cn (S.C.); b1910138@ustb.edu.cn (X.W.)

**Abstract:** More than 500 million people will be added to Africa's cities by 2040, marking the largest urbanization in history. However, nonrenewable fossil energy sources are inadequate to meet Africa's energy needs, and their overexploitation leads to intensified global warming. Fortunately, Africa has a huge potential for biomass energy, which will be an important option for combating climate change and energy shortage. In this study, we present a typical large-scale biogas plant in Burkina Faso, West Africa (Ouagadougou Biogas Plant, OUA), which is the first large-scale biogas generation plant in West Africa. The primary objective of OUA is to treat human feces, and it serves as a demonstration plant for generating electricity for feed-in tariffs. The objectives of this study are to assess the greenhouse gas reduction capacity and economic, environmental, and social benefits of OUA and to analyze the opportunities and challenges of developing biogas projects in Africa. As a result, the net economic profit of the OUA biogas plant is approximately USD 305,000 per year, with an anticipated static payback period of 14.5 years. The OUA plant has the capacity to treat 140,000 tons of human feces and 3000 tons of seasonal mixed organic waste annually, effectively reducing greenhouse gas emissions by 5232.61 tCO$_2$eq, improving the habitat, and providing over 30 local jobs. Finally, the development of biogas projects in Africa includes advantages such as suitable natural conditions, the need for social development, and domestic and international support, as well as challenges in terms of national policies, insufficient funding, technical maintenance, and social culture.

**Keywords:** renewable energy; biomass; biogas plant; greenhouse gas

## 1. Introduction

The overuse of fossil fuels due to rapid industrialization and urbanization is accelerating global warming. In addition, the increase in population and rapid economic and industrial development has led to a massive increase in global solid waste production [1,2]. Municipal solid waste (MSW) is a major contributor to climate change, responsible for over 70% of global greenhouse gas (GHG) emissions [3,4]. Therefore, finding alternative energy sources and advanced energy use technologies is crucial in reducing dependence on fossil fuels, GHG emissions, and the greenhouse effect without negatively affecting population and economic growth [5–7]. Studies have shown that the development of renewable energy and modern energy utilization technologies is an effective solution to

the problems of climate change, energy stress, and waste management [8–10]. Biogas projects, as a form of renewable energy, have great potential in the development of modern energy utilization [11]. The use of effective biogas engineering techniques can minimize the impacts associated with global warming and climate change while improving the human environment [4].

In Africa, around 600 million people have no way to access electricity, and around 700 million use non-clean cooking [12]. The shortage of electricity severely limits social and economic development in Africa [13]. The International Energy Agency (IEA) predicts that Africa's total electricity demand will grow at an average rate of 4% per year by 2040. However, even by 2030, some 500 million people will still lack access to electricity due to population growth and other factors [14,15]. Energy poverty is defined as a lack of access to electricity and a heavy reliance on traditional biomass. It is widespread and unevenly distributed in Africa, with the most severe cases found in West Africa [16]. "Traditional uses of biomass" refers mainly to the inefficient use of solid biomass by low-income households that do not have access to modern energy sources and technologies [17]. With the exception of South Africa, about 80% of the total primary energy demand in sub-Saharan Africa (SSA) is provided by solid biofuels and biomass feedstocks are mainly used in traditional forms in SSA [18]. Approximately 50% of the total energy used in Africa comes from fuelwood [19]. Unsustainable fuelwood harvesting causes forest depletion, and time-consuming wood collection processes result in lost production time and place a heavy burden on human health, particularly on women and children [12,20]. Fossil fuels account for approximately 40.0% of the overall energy mix in SSA, with coal accounting for 13.0% and natural gas for 16.0% [21]. More than 80% of urban households in SSA use charcoal as a cooking fuel [22].

The incomplete combustion of solid fuel produces large amounts of GHGs and particulate matter, resulting in the accumulation of indoor smoke and air pollution, which is hazardous to human health [23–25]. The World Health Organization estimates that 7 million people die each year from diseases related to environmental and household air pollution [26]. Studies have shown that air pollution caused 1.1 million deaths across Africa in 2019, of which 697,000 were caused by indoor air pollution, and 394,000 were caused by ambient air pollution [27]. In recent years, as urbanization in Africa has accelerated, SSA, particularly in West Africa, has become heavily dependent on unsustainable energy sources for economic growth, leading to a dramatic increase in carbon emissions [28]. In addition to energy issues, Africa's poor infrastructure, especially inadequate sanitation systems, poses a huge challenge to human health. Over 80% of the population in SSA use unimproved on-site sanitation facilities (toilets not connected to sewers) [29]. Untreated fecal matter exposed to the air produces unpleasant odors and breeds bacteria, increasing the pathway for the spread of germs and posing a serious threat to human health [30].

Biogas technology is one of the modern bioenergy utilization technologies that has been widely concerned by all countries in the world due to its obvious advantages over other renewable energy utilization technologies in terms of waste management, environmental sanitation, human health, and energy utilization [31]. Europe is the global leader in biogas generation, with 18,943 biogas projects established as of 2019, accounting for 65% of global biogas generation capacity, which is about 21.6 GW [32]. Germany is the world's number one producer of biogas, accounting for half of Europe's biogas production, with advanced biogas production technology leading to the development of biogas plants worldwide. Germany was one of the first European countries to introduce subsidies for renewable electricity and biogas production, and the introduction of the Renewable Energy Act (Erneuerbare Energien Gesetz) in 2000 has effectively accelerated the development of biogas plants in the country. By the end of 2019, 9527 biogas plants had been developed in Germany [33]. However, the eco-friendliness of energy crops, the second most utilized feedstock in European biogas plants, for biogas production is questioned because of the impact on soil fertility and food production [34,35].

Biogas projects are more widely used in rural areas of developing countries and are an integral part of securing agriculture, waste management, and energy security, mainly

represented by small biogas digesters. A total of approximately 50 million small-scale digesters are in operation worldwide, mainly in China and India. In 2017, China produced 12.366 $Gm^3$ of biogas, capable of replacing around 8.605 Mt of standard coal per year [36,37]. Estimations show that 700,000 biogas plants have been installed in other parts of Asia, Africa, and South America [38]. Although the number of large-scale biogas plants in Africa is currently small, the potential to be exploited is huge [39]. Tumwesige et al. highlighted the huge potential for biogas use in rural areas of SSA [40]. According to the International Renewable Energy Agency (IRENA), biogas production has increased remarkably over the last decade. The global distribution of total biogas production for electricity generation at the end of 2020 is shown in Figure 1 [41].

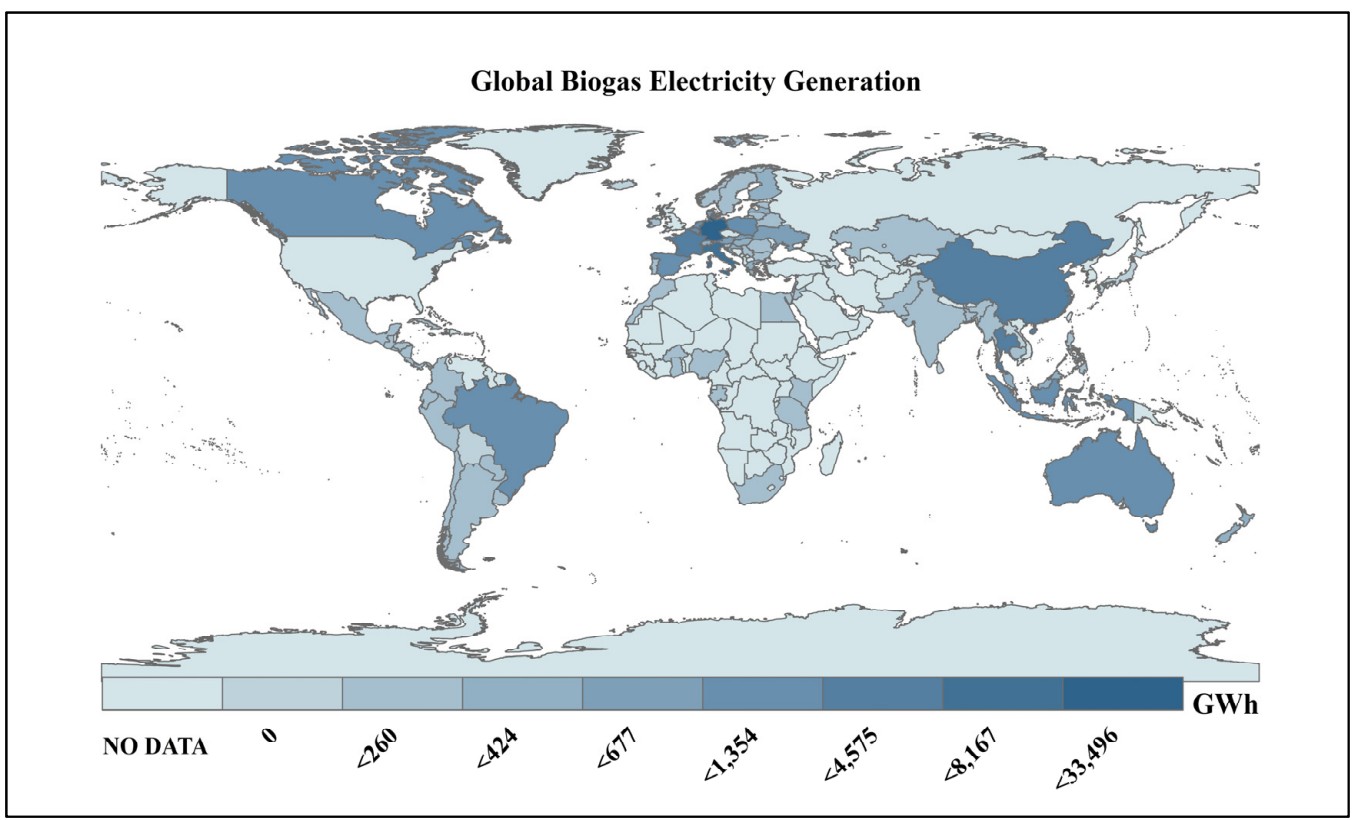

**Figure 1.** Global distribution of biogas generation capacity in 2020. Source: IRENA, Renewable Energy Statistics 2022, summarized by the authors.

The Ouagadougou biogas plant (OUA) in Burkina Faso, funded by the Bill and Melinda Gates Foundation, is the first large-scale industrial biogas plant in West Africa. Its primary objective is to treat human feces and generate biogas for grid electricity, and it is also the first biogas demonstration plant in West Africa. The feedstock for the OUA biogas plant is mainly human feces and organic waste from neighboring plants, such as jatropha and water hyacinth. This study assesses the carbon reduction capacity of the plant and evaluates its economic, environmental, and social benefits. The opportunities and challenges of developing biogas technology in Africa are explored, particularly in low-income developing countries such as Burkina Faso.

## 2. Materials and Methods

### 2.1. Data Collection

The GHG reduction potential of the OUA biogas plant was calculated by using the 2019 Refinement to the 2006 IPCC guidelines for national greenhouse gas inventories (IPCC) and Clean Development Mechanism (AMS-III. D) [42,43]. Next, a quantitative analysis of the benefits of OUA was conducted by using cost-benefit analysis. The opportunities

and challenges affecting the development of renewable energy sources, such as biogas projects in African countries, were analyzed in the context of the current status of biogas development in Africa.

The data collection procedure for this study involved three main steps.

(1) Designing a questionnaire to be sent to the plant manager of the OUA biogas plant. The questionnaire covered aspects such as initial investment, operation and management, profit from by-products, and related costs of environmental management and treatment.

(2) Reviewing the preparation and preliminary design phase of the OUA biogas plant between 2015 and 2017.

(3) Collecting data on renewable energy sources in Africa, including biogas, from publicly available statistics. The parameters for GHG emission calculations were obtained from the IPCC, the Kyoto Protocol to the United Nations Framework Convention on Climatic Change (UNFCCC) [44] and on-site research conducted by the University of Science and Technology Beijing (USTB) and Chengdu Detong Environmental Engineering Co., Ltd. (Chengdu, China)

*2.2. Study Subjects*

The study focused on Ouagadougou, the capital city of Burkina Faso, which has a catchment area of 51,800 hectares and an estimated population of 2.64 million. The city faces challenges in accessing basic urban services, inadequate housing, unemployment, and urban insecurity. Less than 10% of the population is connected to the central sewerage network, and the majority face on-site problems, such as household toilets and cesspits that require regular maintenance and emptying. The location of the OUA plant is shown in Figure 2. The OUA biogas plant is centered on a large continuous stirred tank reactor (CSTR) with a volume of 2500 m$^3$. The plant uses multistage digestion to produce biogas, which can generate 7000 kWh of electricity per day for use in the grid. The plant processes 400 tons of human feces and 5–10 tons of seasonal mixed organic matter per day, including organic waste, such as jatropha oil press cake, water hyacinth, fruit, and vegetable waste. The main objective of the OUA biogas plant is the co-digestion of human feces and waste organic matter, leading to biogas production, GHG reduction, and environmental improvement. The main facilities of the biogas plant include a CSTR, cover anaerobic lagoon, gasholder, and combined heat and power (CHP) units. The specific construction facilities are shown in Table 1. The process flow, mainly composed of raw material pretreatment, hydrolysis acidification, anaerobic digestion, desulphurization, grid, and production of biogas fertilizer, is shown in Figure 3.

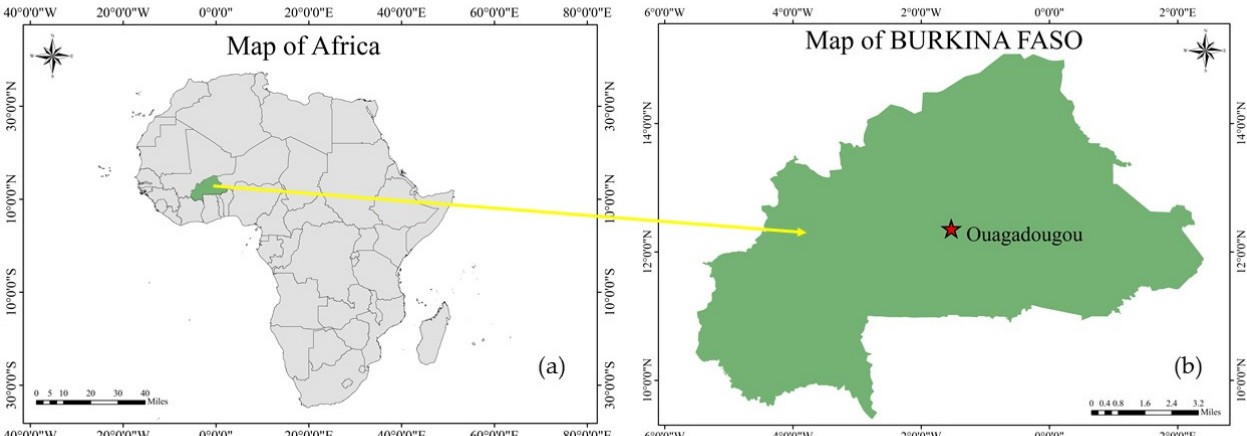

**Figure 2.** Geographical coordinates of Burkina Faso (**a**) and geographical coordinates of the OUA biogas plant (**b**).

**Table 1.** Main facilities of the OUA biogas plant.

| Nr. | Structure | Amount | Parameter |
|-----|-----------|--------|-----------|
| 1 | Sedimentation pound | 1 | 600 m$^3$ |
| 2 | Acidification tank | 1 | 300 m$^3$ |
| 3 | Pasteurization tank | 1 | 100 m$^3$ |
| 4 | Adjustment pound | 1 | 45 m$^3$ |
| 5 | CSTR tank | 1 | 2500 m$^3$ |
| 6 | Lagoon | 1 | 1300 m$^3$ |
| 7 | Buffer tank | 1 | 100 m$^3$ |
| 8 | Gasholder | 1 | 1500 m$^3$ |
| 9 | CHP units | 2 | 1.1 MW |
| 10 | Solid separator | 1 | 45 m$^3$/h |
| 11 | Torch | 1 | 150 m$^3$/h |

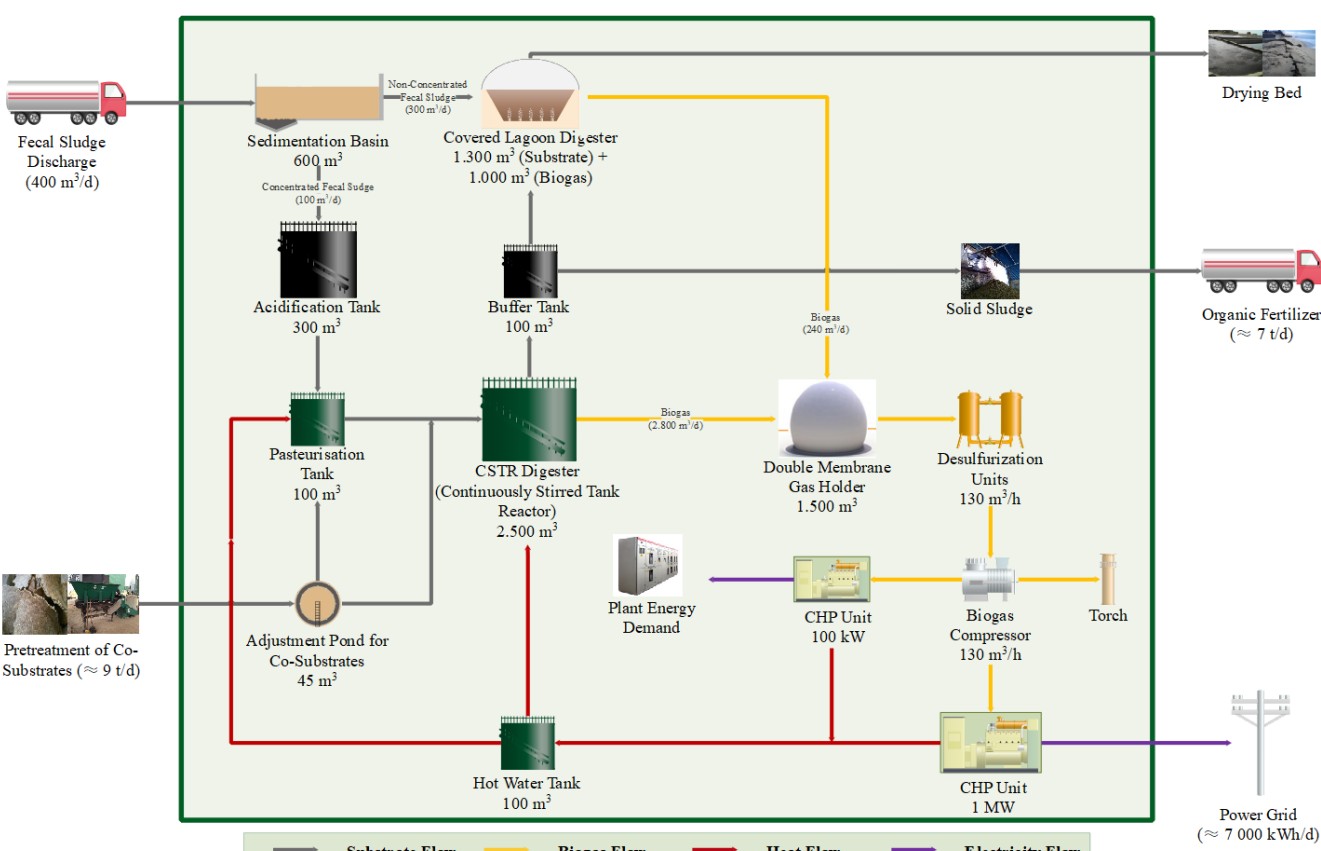

**Figure 3.** Operational process flow of the OUA biogas plant.

### 2.2.1. Raw Material Pretreatment Systems

Considering that the biogas content produced by anaerobic fermentation using fecal sludge alone is extremely low, mixed digestion is beneficial for improving the stability and productivity of the process [45]. As part of this process, 100 tons of concentrated human fecal sludge are extracted from 400 tons of fecal effluent per day, along with 5–10 tons of co-fermented materials, such as jatropha press cake and brewery waste. The sand in the incoming fecal sludge is first separated and removed by using a screw desander to prevent problems, such as blockages in the material transfer pipeline and deposition in the fermentation tank.

### 2.2.2. Hydrolysis and Acidification Process

This process reduces the possibility of a dangerous anaerobic tank acidification accident. Micro-organic matter in the hydrolysis is broken down to improve the homogeneity of the substrate and reduce odor and carbon dioxide emissions. A small amount of carbon dioxide is yielded in this stage, leading to the rise of final methane content in the produced biogas. The acidification step takes approximately 3 days, after which the substrate is pasteurized at 70 °C for 1 h to eliminate pathogens and bacteria and make the fermentation residue safe for use as a biofertilizer. The substrate is then left in the pasteurization tank, and the temperature is reduced before being pumped into the fermenter.

### 2.2.3. Fermentation, Biogas Purification, and Desulfurization Technology

The methanization step is divided into two parts: a CSTR and a covered anaerobic lagoon. The substrate is reacted in the CSTR with a volume of 2500 m$^3$ at 38 °C for approximately 20 days. Subsequently, the digestate passes through the buffer tank and solid–liquid separator. The solid digestate can be collected and applied as a biofertilizer. The liquid part flows into the covered lagoon digester before being directed to the neighboring drying bed. In the two units, CSTR digester and covered lagoon digester, about 3000 m$^3$ of biogas will be produced per day in full operation.

The first desulfurization step involves adding FeCl$_2$ liquor during the acidification process. The chemical redox reaction between ferrous ions and sulfide decreases the formation of other sulfide compounds, such as hydrogen sulfide. Therefore, corrosion damage to plant components is effectively prevented, ensuring better methane production in the fermentation step. The second desulfurization step consists of an active carbon filter unit to minimize the hydrogen sulfide concentration and protect the CHP unit, guaranteeing durability. The purified gas is stored in a double membrane gas cabinet with a volume of 1500 m$^3$. The above stabilization process allows for effective bioenergy recovery and the conversion of more than 80% of volatile solids into biogas.

### 2.2.4. Combined Heat and Power Generation

The biogas is used to generate electricity via two CHP units with a total installed capacity of 1100 kW. One of the CHP units (100 kW) covers the energy demand of the plant equipment and enables self-sufficiency of the biogas plant. The larger CHP unit (1 MW) generates 7000 kWh per day and feeds electricity to the power grid. In addition, the 445 kW heat loss from the 1 MW cogeneration is used to preheat the substrate in the pasteurization tank or to heat the CSTR tank, which can heat the liquid in the pasteurization tank to 70 degrees centigrade and maintain that temperature for at least 1 h.

### *2.3. GHG Emission Reduction Calculation*

The GHG emission reduction assessment was conducted on the basis of the baselines and plant emissions [46]. The baseline emissions were calculated under scenarios where human feces and seasonal mixed organic matter, such as jatropha press cake, are used as co-substrates. The first scenario was that human feces was left to decay anaerobically within the plant boundary, and methane was emitted directly into the atmosphere. The second scenario was that the seasonal mixed organic matter was left to decay at the solid waste disposal site (SWDS). The GHG emission reductions from the OUA biogas plant and GHG emissions from the baseline scenario are calculated by Equation (1) and Equation (2), respectively [42,46].

$$ER_{CH_4} = BE_{CH_4} - PE_{OUA} \tag{1}$$

where

$ER_{CH4}$ = GHG emission reduction from OUA biogas plant in year (t CO$_2$ eq);
$BE_{CH4}$ = Baseline scenario GHG emissions in year (t CO$_2$ eq);
$PE_{OUA}$ = Project activity emissions in year (t CO$_2$ eq);

$$BE_{CH_4} = BE_{HF} + BE_{SWDS} \tag{2}$$

where

$BE_{HF}$ = Baseline emission of human feces in the open lagoon scenario in year (t $CO_2$ eq);

$BE_{SWDS}$ = Baseline emissions of seasonal mixing of solid waste in the SWDS scenario in year (t $CO_2$ eq);

### 2.3.1. Open Anaerobic Pond Scenario in an Open Lagoon, the Baseline Discharge of Human Feces Is Calculated as Follows

The OUA biogas plant treated 400 tons of human feces every day. If these human feces are not properly treated and disposed of, then they will be easy to breed bacteria, emit a foul odor, pollute the natural ecological environment, and seriously endanger human health [47]. In addition, the GHGs produced will be emitted directly into the atmosphere, causing negative impacts on global warming and climate change [4]. Thus, the biogas plant plays an important role in the GHG emission process. The baseline emission of human feces in an open lagoon is calculated as follows [42,43]:

$$BE_{HF} = VS_{HF} \times B_O \times D_{CH_4} \times UF_b \times MCF_{HF} \times GWP_{CH_4} \tag{3}$$

where

$VS_{HF}$ = Total organic matter of human feces used in an anaerobic co-digestion plant in year (919,800 kg in this study).

$B_o$ = Maximum methane production potential of the volatile solid ($m^3$ $CH_4$/kg VS) (0.35 $m^3$ $CH_4$/kg VS, according to onsite study by USTB and DeTong Knowledge);

$D_{CH4}$ = $CH_4$ density (0.67 kg/$m^3$ at 20 °C and 1 atm pressure);

$UF_b$ = Model uncertainty correction factor (0.94 recommended by the FCCC);

$MCF_{HF}$ = Annual methane conversion factor (80.0%, recommended by the IPCC);

$GWP_{CH4}$ = Impact of CH4 relative to $CO_2$ on global warming potential (t $CO_2$/t $CH_4$) (28 from the IPCC [48]).

### 2.3.2. Solid Waste Disposal Site

Seasonal mixed fermentation feedstocks include organic waste, such as jatropha and water hyacinth, which are mainly derived from surrounding industries, agriculture, and plantations. Organic waste that has not been effectively treated can be a waste of resources and a burden on the natural environment. On the basis of the first-order decay method (FOD), the baseline emission of seasonal mixed organic waste at solid waste disposal sites is determined as follows [42,43]:

$$BE_{SWDS} = DOC_f \times MCF \times M \times DOC \times \frac{16}{12} \times F \times \left(1 - e^{-k}\right) \times (1 - OX) \times (1 - f) \times \varphi \times GWP_{CH_4} \tag{4}$$

where

$DOC_f$ = Fraction of degradable organic carbon degraded and released in SWDS for year (0.5, recommended by the IPCC);

$MCF$ = Methane correction factor for unmanaged SWDS (80.0%, recommended by the IPCC);

$M$ = Amount of seasonal mixing of solid waste disposed of in SWDS in year (2051.3 t/year in this study);

$DOC$ = Fraction of degradable organic carbon in seasonal mixing of solid waste (15.0%, recommended by the IPCC);

$F$ = Fraction of methane in SWDS gas (0.5, recommended by the IPCC);

$k$ = Decay rate for seasonal mixing of solid waste (1/year) (0.05, recommended by the IPCC);

$OX$ = Oxidation factor (0.1, recommended by the IPCC);

$f$ = Fraction of methane captured and treated, burned, or otherwise used to prevent the release of methane into the atmosphere in year (0.5, recommended by the UNFCCC);

$\varphi$ = Model correction factor (0.8, recommended by the IPCC).

### 2.3.3. Plant Activity Emissions

Project emissions associated with the OUA biogas plant are determined by Equation (5). The GHG emissions by biogas leakage are calculated by Equation (6). The annual GHG emission reduction from electricity generation using produced methane as a substitute for coal is calculated through Equation (7) [42,43].

$$PE_{OUA} = PE_{EC} + PE_{CH_4} - AG_{CH_4} \tag{5}$$

where

$PE_{EC}$ = Project activity emissions from electricity consumption in year (t $CO_2$ eq);
$PE_{CH4}$ = Project emissions of methane leakage in year (t $CO_2$ eq);
$AG_{CH4}$ = GHG emission reductions from methane replacement of coal for power generation in year (t $CO_2$ eq);

$$PE_{CH_4} = Q_{Biogas} \times D_{CH_4} \times f \times EF_{CH_4} \times GWP_{CH_4} \tag{6}$$

where

$PE_{CH4}$ = Project emissions of methane leakage in year (t $CO_2$ eq);
$Q_{Biogas}$ = Quantity of biogas produced in the digester in year (1,080,000 $m^3$ biogas)
$f$ = Value for a fraction of methane in the biogas (default, 60% $m^3$ $CH_4/m^3$ biogas)
$EF_{CH4}$ = Emission factor for a fraction of $CH_4$ produced that leaks (default, 10% fraction)

$$AG_{CH_4} = EF_{coal} \times Q \times D_{CH_4} \times \frac{NCV_{CH_4}}{NCV_{coal}} \tag{7}$$

where

$AG_{CH4}$ = GHG emissions reduced by methane captured and effectively used by the plant activity in year (t $CO_2$ eq);
$EF_{coal}$ = GHG emissions from the use of standard coal (2.658, recommended by the IPCC);
$Q$ = Annual methane production from biogas digester (648,000 $m^3$ $CH_4$);
$NCV_{CH4}$ = Net calorific value of methane with a default value of 50.4 MJ/kg;
$NCV_{coal}$ = Net calorific value of standard coal with a default value of 29.307 MJ/kg.

## 3. Results and Discussion

### 3.1. GHG Potential Reduction

The OUA biogas plant reduces GHG emissions by 5232.61 t$CO_2$eq per year through co-digestion of human feces and seasonal mixed organic waste; the specific GHG emission calculation results are shown in Table 2. Some of the literature on GHG emission reductions from biogas projects can be found in Table 3. Family-scale biogas projects mainly use animal manure as the main fermentation material to produce biogas for electricity and heat. Farm-scale biogas projects using agricultural waste as fermentation raw material are generally medium to large in size, with a daily biogas production greater than 150 $m^3$. The main fermentation raw materials include livestock and poultry manure, such as cow dung and pig manure, and agricultural waste, such as maize straw. Large-scale biogas projects have relatively higher biogas production and show good GHG emission reduction capacity by utilizing various agricultural waste resources.

In anaerobic systems, the co-digestion of organic waste can improve system stability and gas production efficiency, contributing to GHG reduction [49]. The OUA biogas generation plant is the first large-scale biogas plant in West Africa with the main objective of treating human feces and producing 1,080,000 $m^3$ of biogas per year, with a volumetric biogas production rate of 1.18 $m^3/m^3$ per day. Compared to the human feces biogas plant in Cui Ge Zhuang village, China (where the feedstock is almost exclusively human feces), the OUA biogas plant has a higher volumetric biogas production rate [50]. The plant effectively utilizes human feces and seasonal organic waste, such as jatropha press cake,

for co-digestion, achieving harmless treatment and resourceful use of all types of organic waste while reducing GHG emissions and generating environmental benefits.

**Table 2.** Baseline variables for carbon emission calculations.

| Variable | Description | Data (tCO$_2$eq/Year) |
|---|---|---|
| BE$_{HF}$ | Baseline emission of human feces in the open lagoon scenario in year | 4541.63 |
| BE$_{SWDS}$ | Baseline emissions of seasonal mixing of solid waste in the SWDS scenario in year | 40.34 |
| PE$_{EC}$ | Project activity emissions from electricity consumption in year | 118.27 [a] |
| PE$_{CH4}$ | Project emissions of methane leakage in year | 1215.65 |
| AG$_{CH4}$ | GHG emission reductions from methane replacement of coal for power generation in year | 1984.56 |
| BE$_{CH4}$ | Baseline scenario GHG emissions in year | 4581.97 |
| PE$_{OUA}$ | Project activity emissions in year | −650.64 |
| ER$_{CH4}$ | GHG emission reduction from OUA biogas plant in year | 5232.61 |

[a] The daily electricity consumption of the OUA biogas plant is 315 kWh, and the consumption of 1 kWh of electricity generates 0.997 kg of CO$_2$ [49].

**Table 3.** Summary of GHG reduction effects of selected biogas plants.

| Substrates | Type | Biogas Production (m$^3$/Year) | Volumetric Biogas Production Rate (m$^3$/(m$^3$·d)) | GHG Emission Reduction (tCO$_2$eq/Year) | Reference |
|---|---|---|---|---|---|
| Food waste | Pilot-scale | 3103 | 4.25 | 0.11 | Liu et al. [51] |
| Cow dung | Family-scale | 355 | 0.48 | 1.40 | Haryanto et al. [52] |
| Cow dung | Family-scale | 578 | 0.26 | 5.29 | Haryanto et al. [53] |
| Kitchen waste and sludge | Industrial-scale | 862,313 | NR | 1554.9 | Guo et al. [49] |
| Cow manure | Farm-scale | 2400 | 0.41 | 0.24 | Richards et al. [54] |
| Pig manure and corn straw | Farm-scale | 20,415 | 1.22 | 303.08 | Wang et al. [55] |
| Pig manure | Farm-scale | 116,800 | 0.40 | 1334.95 | Chen et al. [56] |
| Pig manure | Farm-scale | 321,200 | 0.40 | 4016.95 | Chen et al. [56] |
| Pig manure | Farm-scale | 657,000 | 0.82 | 5236.95 | Chen et al. [56] |
| Straw | Farm-scale | 485,450 | 0.81 | 5582.03 | Wang et al. [57] |
| Pig manure | Farm-scale | 6,570,000 | NR | 49,300 | Zhang et al. [58] |
| Human feces | Industrial-scale | 58,000 | 0.40 | 69.2 | Zhang et al. [50] |
| Human feces | Industrial-scale | 1,080,000 | 1.18 | 5232.61 | This study |

*3.2. Benefits of the OUA Biogas Plant*

3.2.1. Economic Benefits of the OUA Biogas Plant

The initial investment cost of the OUA was approximately USD 4,435,000, with all the items invested listed in Table 4.

The biogas plant currently produces 1,080,000 m$^3$ of biogas and 2.16 million kWh of electricity per year, providing a convenient source of clean electrical energy for the local area and alleviating the current electricity shortage. The direct economic return on electricity is USD 300,000 per year. The OUA biogas plant produces 2500 tons of organic fertilizer each year, providing high-quality fertilizer for local agricultural production, with a direct economic return of USD 25,000 per year from the fertilizer. The environmental treatment costs of the relevant authorities can be reduced by treating organic waste, such as human feces, residues from jatropha press extraction, and fruit and vegetable waste, generating an economic benefit of approximately USD 70,000 per year. Thus, the total economic benefits of the OUA biogas plant can reach approximately USD 395,000 per year.

**Table 4.** Initial investment of biogas plant.

| Nr. | Item | Investment (10$^4$ $) | Percentage |
|---|---|---|---|
| 1 | Manure collection system | 4 | 0.90 |
| 2 | Manure pretreatment system | 10 | 2.25 |
| 3 | Anaerobic digestion system | 100 | 22.55 |
| 4 | Biogas utilization system | 100 | 22.55 |
| 5 | Solid manure/biogas residue (producing organic fertilizer)system | 45 | 10.15 |
| 6 | Subsidiary facilities | 80 | 18.04 |
| 7 | Land use costs | 3 | 0.68 |
| 8 | Public facilities (fire control, roads, landscaping, etc.) | 1.5 | 0.34 |
| 9 | Power supply system | 40 | 9.02 |
| 10 | Other costs (design fee and contingency fee) | 60 | 13.53 |
| | Total | 443.5 | 100 |

The operating costs of the OUA are approximately USD 90,000 per year (Table 5) and consist mainly of maintenance, labor, management, energy consumption, and materials (accessories). After operating costs, the net economic profit of the OUA biogas plant is approximately USD 305,000 per year, with an anticipated static payback period of 14.5 years, not considering reductions in GHG emissions.

**Table 5.** Operating costs for the biogas plant.

| Nr. | Item | Running Cost (10$^4$ USD) | Percentage |
|---|---|---|---|
| 1 | Maintenance costs | 2 | 22.22 |
| 2 | Labor costs | 3 | 33.33 |
| 3 | Management expenses | 1 | 11.11 |
| 4 | Energy consumption costs | 1.5 | 16.67 |
| 5 | Materials (accessories) costs | 1 | 11.11 |
| 6 | Other costs | 0.5 | 5.56 |
| | Total | 9 | 100 |

3.2.2. Environmental and Social Benefits of the OUA Biogas Plant

Burkina Faso is a country with abundant biomass resources, but its efficient use of biomass is limited, and it has only recently started to use biogas for electricity generation, beginning in 2016 (Table 6). According to the IEA, as of 2020, only 21% of the population has access to electricity, and 11% of the population uses clean cooking fuels, making Burkina Faso one of the lowest-ranked countries in West Africa in terms of access to clean energy. The other nine West African countries with less than 5% of the population having access to clean cooking are not shown in Figure 4.

**Table 6.** Bioenergy use in Burkina Faso, 2013–2020.

| Type | | 2013 | 2014 | 2015 | 2016 | 2017 | 2018 | 2019 | 2020 |
|---|---|---|---|---|---|---|---|---|---|
| Installed Capacity (MW) | Liquid Biofuels | 0.15 | 0.15 | 0.15 | 0.15 | 0.15 | 0.15 | 0.15 | 0.15 |
| | Biogas | NR | NR | 0.75 | 0.75 | 0.75 | 0.75 | 0.75 | 0.75 |
| Electricity Generation(GWh) | Liquid Biofuels | 0.05 | 0.15 | 0.30 | 0.30 | 0.30 | 0.30 | 0.30 | 0.30 |
| | Biogas | NR | NR | NR | NR | 1.00 | 1.00 | 1.00 | 1.00 |

Source: IRENA, Select table, 2022, summarized by the authors.

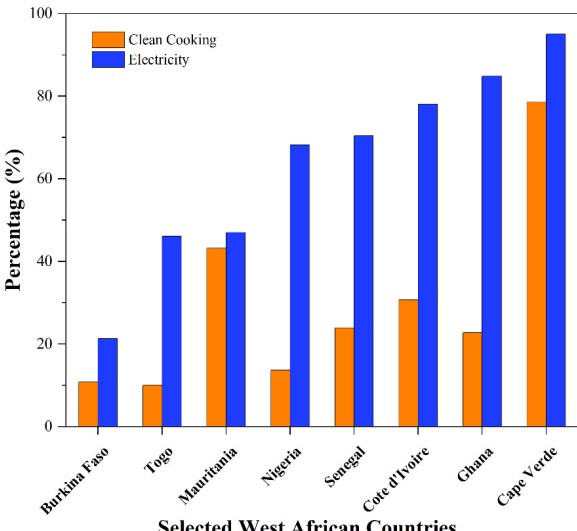

**Figure 4.** Proportion of population with access to electricity and clean cooking in selected African countries in 2020. Source: IEA, SDG7 Database, 2022, summarized by the authors.

The OUA biogas plant treats 140,000 tons of human feces and 3000 tons of seasonal mixed organic waste per year, producing 1,080,000 m$^3$ of biogas, 2,160,000 kWh of electricity, and 2500 tons of organic fertilizer per year, and achieving a GHG reduction of 5232.61 tCO$_2$eq. This reduction has a positive impact on the environment by decreasing pollution, GHG emissions, and the spread of disease. The OUA biogas plant provides appropriate biogas technology for the treatment of sewage and human feces, which can help reduce disease transmission channels, improve the sanitary conditions of the local population's toilets and living environment, and create employment opportunities for over 30 people. Therefore, the OUA plant generates good environmental and social benefits.

The successful construction and operation of the OUA mark the first large-scale commercial human feces biogas power generation plant supported by the Bill & Melinda Gates Foundation, which is of practical importance in achieving the objectives of the strategic cooperation between the Chinese Ministry of Science and Technology and the Gates Foundation.

## 4. Opportunities and Challenges

### 4.1. Main Opportunities for Biogas Project Development in Africa

#### 4.1.1. Favorable Natural Conditions

Africa has a typical hot, low rainfall and dry climate [59]. Burkina Faso has a savannah climate with an average annual temperature of around 27 °C. The inherent climatic advantage makes the additional energy required to insulate anaerobic digesters much less, creating favorable conditions for the development of biogas projects in the region. Approximately 630 million hectares of land in SSA are covered by forest, which accounts for about a quarter of the land area [60]. Africa has a huge potential for renewable energy development, and biomass is an important renewable energy source in SSA, including solid resources such as wood, animal manure, and agricultural waste [61]. Biomass energy can provide a large proportion of the grid balance required for a fully renewable power system [62]. It also contributes to the achievement of the sustainable development goals (SDGs), which mainly include improved environmental health (SDG 6), access to affordable, sustainable and stable clean energy (SDG 7), and climate action (SDG 13) [63].

#### 4.1.2. Economic Development and Environmental Health Needs

Currently, for every 1 percentage point increase in per capita electricity consumption on the African continent, its GDP per capita will increase by 0.09 percentage points, highlighting the critical role of electricity in driving economic development [64]. However, the

SSA region still lags behind with only 46.8% access to electricity, leaving 570 million people without electricity, which accounts for three-quarters of the global population without access to electricity [65]. The ongoing COVID-19 epidemic has further exacerbated the situation, with 25% of the region's health facilities lacking electricity and over 70% without access to stable power, hindering the region's efforts to combat the epidemic and promote recovery [66]. Basic sanitation facilities are also not guaranteed in Africa, with 709 million people lacking access to basic sanitation services, and open defecation remains a problem in many areas, such as Burkina Faso, with 9.8 million people affected [29]. In this context, biogas projects offer a promising solution to address the regions' energy and sanitation needs [11]. The anaerobic digestion process generates biogas for electricity generation while also inhibiting potential pathways for the spread of germs from urban waste, especially human feces, and providing basic sanitation services to local communities. In addition, the rational use of biogas energy as an alternative to traditional solid biomass combustion for domestic production can help reduce GHG emissions, such as particulate matter, carbon dioxide, and methane, thereby reducing the greenhouse effect [4].

### 4.1.3. Urbanization and Energy Needs

Despite having the lowest modern energy supply. Africa has the highest urbanization and population growth rates in the world [67]. By 2040, half of the world's new population is expected to be African, with 70% of the growth concentrated in urban areas, and the urban population of Africa will increase by more than 500 million people, making the largest urbanization in history [12]. African countries are currently lagging behind in economic development, which has led to problems such as poor infrastructure, power shortages, and energy shortages. With the acceleration of urbanization, Africa's energy needs will continue to grow, and energy will become central and critical to national development [68]. The development of biogas technology is expected to address the contradiction between Africa's rapid urbanization, energy shortages, and climate change. This condition can contribute to the reducing deforestation in SSA, the pressure on women and children to collect firewood over long distances to meet household needs, and the number of premature deaths caused by air pollution [69–71]. Subedi et al. suggested that biogas from anaerobic digesters can help reduce deforestation on the African continent by up to 26% by 2030 as a result of replacing a portion of firewood consumption with biogas [72]. The development of biogas technology in Africa is important in terms of improving the efficiency of biomass fuels, disposing of increasing amounts of agricultural waste, addressing the threat of urban waste, and improving ecological sanitation [68].

### 4.1.4. Domestic and International Support

In September 2015, world leaders at the UN Summit adopted the 2030 Agenda for Sustainable Development, which aims to eradicate all forms of poverty through 17 SDGs. The seventh SDG aims to achieve universal access to energy and states that "ensuring access to modern, affordable, reliable, and sustainable energy for all" is crucial [63]. The development of large-scale biogas projects is an important measure to achieve sustainable development goals. The OUA biogas plant is the first pilot plant in West Africa, using biogas technology and models from China, funded by the Bill & Melinda Gates Foundation. The plant has received high recognition from the World Bank, the West African Development Bank, the Government of Burkina Faso, and ECOWAS [21,73,74].

### 4.2. Main Barriers to Biogas Project Development in Africa

Africa lacks effective policies for the development of renewable energy. Current African national policies remain biased toward fossil fuels, and the implementation of policies on renewable energy cannot be guaranteed [75,76]. The use of and investment in renewable energy in SSA countries are inhibited by national policies. Inadequate regulations, institutions, and unenforced regulatory authorities have created many obstacles to private investment and biogas projects [77,78]. Especially in rural areas, unstable incomes result in

many households not being able to afford the initial construction costs of a digester, while state subsidies for fossil fuels and the easy availability of traditional biomass fuels make renewable energy uncompetitive. The high initial investment costs for the construction of large-scale biogas projects in Africa and the funding gap caused by the high initial investment costs are inhibiting project financing in the renewable energy sector [79–81].

Secondly, the development of biogas technology in Africa is in its infancy, and equipment and technology for the construction of biogas plants need to be imported from abroad. For example, the most common design in East Africa is the fixed dome Chinese digester [82,83]. In most cases, biogas installers do not provide adequate technical support and sound post-maintenance training to users. The lack of post-maintenance results in biogas systems being prone to breakdowns and damage during operation, which eventually leads to the abandonment of the biogas system [84]. In addition, the design and installation of many biogas projects ignore the needs of the users and local characteristics, such as the seasonality of the feedstock, the quantity of feedstock, and the difficulty of collection, all of which make it difficult to operate the installed biogas projects [84,85]. In addition to the technical challenges, the development of biogas projects in Africa has also been affected by socio-cultural influences. Compared with solar and wind energy, biogas technology is socially unacceptable because it is mainly based on organic pollutants, such as animal manure, agricultural waste, food waste, and toilet waste [75,86]. In addition, the use of biogas for heating, cooking, and lighting is likely to cause physical disgust and moral aggression among the population [75]. Finally, the lagging technical standards and norms, differences in living habits, and political security issues also make it difficult to develop biogas projects in Africa to varying degrees [87,88]. Attracting investors to biogas projects is difficult in areas where security is not available because biogas projects are long-term investments [82].

Although the development of renewable energy has received attention in Africa, the continent's share of renewable energy in total final energy consumption was still on a declining trend [89]. A causal loop diagram using system dynamics (Figure 5) reveals that active national policies, appropriate subsidies, and technical expertise are particularly important in promoting biogas projects in Africa.

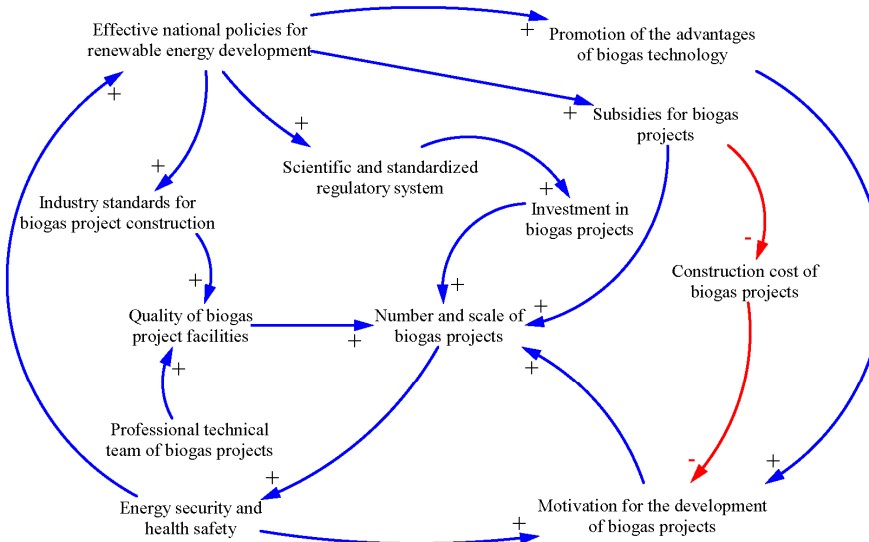

**Figure 5.** The main factors affecting the development of biogas projects in Africa. Note: The line with arrows connecting two variables is called a causal chain. Causal loop "+" indicates that the variable to which the arrow points will increase (or decrease) with the increase (or decrease) of the variable from which the arrow originates; Causal loop "-" indicates that the variable to which the arrow points will decrease (or increase) with the increase (or decrease) of the variable from which the arrow originates.

## 5. Conclusions and Recommendations

The overuse of fossil fuels is causing increased GHG emissions, and biogas technology has proven to be a successful means for many nations to battle climate change, reduce GHG emissions, conserve forest vegetation, and secure energy sources, among other needs. The OUA biogas plant in Burkina Faso aims to produce biogas for electricity generation by effectively using human feces and seasonal organic waste. It is the first industrial-scale biogas plant in West Africa with a major focus on treating human feces. The scaled-up biogas plant creates a closed-loop system from waste collection to treatment to use, accounting for energy generation and waste management, which can lessen reliance on fossil fuels and encourage energy independence in the African region. The carbon emission reduction analysis and cost-benefit analysis show that the OUA plant has brought good social, environmental, and economic benefits and inspired African countries and other low and middle-income countries to expedite their energy transformation. However, the OUA biogas plant is supported by funds from the Bill Gates Foundation, and the main purpose of building the OUA biogas plant is to solve the sanitation problems of the local population. Therefore, trying to commercialize such a biogas plant in Africa is difficult to achieve due to financial and other problems.

Specifically, the OUA biogas plant in Burkina Faso has good economic benefits, with a direct economic return of USD 325,000 per year through electricity generation and the organic fertilizer produced. The Ouagadougou, Burkina Faso, biogas facility also offers major environmental and social advantages. The biogas plant can process 140,000 tons of human feces and 3000 tons of seasonal organic waste annually, resulting in a 35232.61 $tCO_2eq$ reduction in GHG emissions, mitigating environmental pollution, lowering the risk of germ transmission, and providing the community with a clean, healthy environment to live in. At the same time, it can simultaneously support local employment and provide jobs for more than 30 people. Finally, the development of biogas projects in Africa includes advantages such as suitable natural conditions, the need for social development, and domestic and international support, as well as challenges in terms of national policies, insufficient funding, technical maintenance, and social culture.

A better balance between faster urbanization and rising energy use needs to be achieved in African nations. Although establishing biogas projects in Africa has many advantages, considering potential drawbacks, such as ineffective national policies and regulations, high start-up costs, and antiquated technology, is essential. To address these issues, we recommend that African governments create incentives for renewable energy projects, provide energy subsidies to renewable energy companies to promote increased production, and involve the public and private sectors in expanding energy financing. We also suggest promoting policies that attract foreign direct investment, raising awareness about the advantages of using biogas technology in communities, and training professional technical maintenance teams. The success of the OUA biogas plant highlights the need for increased financial and technical support for additional biogas projects and the development of cutting-edge energy use technologies that promote economic growth while reducing GHG emissions and minimizing environmental and public health risks.

**Author Contributions:** Conceptualization, H.C., Q.X. and S.C.; data curation, T.W. and T.B.; software, H.C. and X.W.; writing—original draft preparation, H.C. and S.C.; writing—review and editing, X.W., S.P.L. and H.-P.M.; supervision, Z.L.; project administration, S.C. All authors have read and agreed to the published version of the manuscript.

**Funding:** This study was funded by the National Key Research and Development Plan (2018YFC1903206), Natural Science Foundation of China (52261145693).

**Institutional Review Board Statement:** Not applicable.

**Informed Consent Statement:** Not applicable.

**Data Availability Statement:** Not applicable.

**Acknowledgments:** The authors would like to take this opportunity to express our sincere appreciation for the support of Office national de l'eau et de l'assainissement (ONEA) in Burkina Faso, Bill & Melinda Gates Foundation, China National Environment, and Energy International Science and Technology Cooperation Base.

**Conflicts of Interest:** The authors declare no conflict of interest.

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
