# Peer review of "Comprehensive Analysis and Greenhouse Gas Reduction Assessment of the First Large-Scale Biogas Generation Plant in West Africa"

_atmosphere, doi:10.3390/atmos14050876_

Round 1

Reviewer 1 Report

Highlight changes in yellow in a next revision, please. No track changes.

Authors starting by the supplementary file:

It contains some similarity. And I see no reference being mentioned when that happens, in the items, for example. Also, it makes no sense to present percentage above in the heading, and then the symbol of percentage in every value. This is just not done like this.

Then, if the supplementary material contains references, the list of references used, if it is a separated document, needs to be present here.

Now let’s go back to the main document. What happens is that similarity is also present in several places. The document being used is the same and references differ or are missing, in many cases.

There is no need to present similarity at all, unless the authors are using some variables, names, Etcetera. And they need to justify that.

Global comments then:

I would suggest to be careful with the language used. Is there any ways that is not human?!

“t human waste a”

Please revise the international unit system.

I would like to see clear findings and practical implications. These are mostly objectives.

“This study presents a comprehensive analysis of the economic, environmental, and social benefits of OUA, and highlights the advantages of developing biogas projects in Africa. It also em- phasizes the challenges facing the development of biogas projects in Africa, such as insufficient funding, technical maintenance, sociocultural aspects, and national policies.”

Then again in the conclusions, please start by brief contextualization and methodology. Clear main findings, practical implications, limitations and future prospects.

The quality of the figure is low, please assure that every figure being presented is original because of the way authors present this figure. I’m not sure it is original or where the data came from. It needs to be present at the end of the caption.

“Figure 1. Global distribution of biogas generation capacity in 2020.”

Why use all these headings one after the other? Why start using the listing style?

“2. Materials and Methods 2.1. Data and Methods 2.1.1. Data Collection The data collection procedure for this study involved three main steps. (1)”

The language used needs to be coherent fluid and informative.

I strongly suggest the authors to avoid the lists…

And again the quality of the figures need to be improved, please. Refer to the comments previously mentioned. And if these figures are too, in fact, they are grouped, then a corresponding subcaption must correspond to each other by letter, after the main caption.

“Figure 2. Geographical coordinates of the OUA biogas plant.”

No need to have similarity here, otherwise just refer to the original document.

“2.2.2. Hydrolysis and Acidification Process The hydrolyzer is a semi anaerobic reactor loaded with a mixture of fecal sludge and pretreated co-substrates, as well as effluent from the buffer tank if necessary. Hydrolysis of polymers and acidification of compounds occur in the first step, yielding soluble small organic matters, such as glucose and volatile fatty acids.”

The entire paragraph contains similarity.

“3.1. GHG Reduction Assessment”

No reference is being mentioned immediately before the equation is presented. However, this is not new.

“causing negative impacts on global warming and climate change. Thus, the biogas project plays an important role in the GHG emission process. The baseline emission of human waste in an open lagoon was approximately 4541.63 tCO2eq per year, calculated as fol- lows:”

Same here and please do not use abbreviations in headings.

“3.1.2. SWDS”

It’s time to term “reference” is used in a columns table, then the names of the authors need to proceed the reference number. It is. It is, in fact, a direct citation.

Strong similarity here.

“3.2. Economic, Environmental and Social Benefits of the OUA Biogas Project 3.2.1. Economic Benefits of the OUA Biogas Project”

Authors are using previous study data to build this one. Then it needs to be highlighted, justified, explained and clarified.

The thought the source is not obvious it is it contains no ear it needs to refer to a URL if that is the case and presented at the final list.

“Source: IRENA, summarized by the authors.”

Differ…

“4. Opportunities and Challenges 4.1. Opportunities 4.1.1.”

Such statement is an example of a statement that needs to be referenced. There is similarity and no source is presented.

“three quarters of the global population without access to electricity”

It is my perspective that the results and discussion section are just to separated a lot of Headings, tiny content in each one.

Despite finding the article important specifically in the contest of developing country, please assure that the above Commons can be used to improve it. Tables are not that useful if the content is not better presented. Consider including some diagrams to refer to the parameters relation, for example, or to the challenges and opportunities being presented.

In terms of references, I would like to see more references from 2023, for example, I do not refer to URLs.

Needs refinement

Reviewer 2 Report

The manuscript entitled „Comprehensive analysis and greenhouse gas reduction assessment of the first large-scale biogas generation project in West Africa”  presents an analysis of the economic, environmental, and social benefits of a large-scale biogas project in Burkina Faso.

The major remarks are as follows:

1.        In Introduction I miss the disadvantage of biogas plants fed with energy crops which are very popular in Europe. Especially the industrial big biogas plants are based on maize silage which is no longer considered as eco-friendly solution. So I suggest to emphasize that the described project is based on wastes not on energy crops which is very valuable issue.

2.        There is also the 2019 Refinement to the 2006 IPCC Guidelines for National Greenhouse Gas Inventories, which should be used in conjunction with the 2006 IPCC Guidelines. Please check it and use it if necessary and add to the references.

3.        The term “project” is somewhat misleading. Please emphasize that this biogas plant is a real working facility. Please consider the term “biogas plant” instead of project.

4.        Could you be more specific and give the information about the sources of feedstock for the biogas plant. I understand that feces are from household toilets and cesspits. Where from are the co-substrates?

5.        From what percentage of the population or households in this city are the human feces delivered to the biogas plant?

6.        How many households (or percentage of households in the city) can be powered by electricity from a biogas plant?

7.        Please describe what happens to the heat from the second larger CHP unit (1 206 MW)? Has this been included in the analysis?

8.        Please describe what happens with digestate. Could you provide the information about its nutrient content?

9.        The first paragraph of the Results and Discussion is rather the Conclusion, not the result or discussion.

10.     The Section 3.1 GHG Reduction assessment is mainly a method of GHG reduction assessment and should be in Section 2. In Results only the results should be shown with the discussion.

11.     The boundaries and the scenarios should be better explained and the results for all scenarios should be shown and compared. From the further text it seems that these are not scenarios but calculations of GHG emissions from typical utilization of human feces and organic wastes. Please be more specific and describe better what emissions were calculated.

12.     The baseline scenario should consider no biogas plant and other scenarios should consider the wastewater treatment plant for human excrements and composting facility for organic waste, perhaps that is what happens to the organic waste. Typically, in agriculture the organic waste is composted. Since there is no information how solid digestate is treated it is difficult to compare the GHG emissions.   

13.     Why the internal GHG emissions were not included in the calculations?

14.     Paragraph in lines 291-305 need the re-writing since it gives the contradictory conclusions. The costs and building materials has nothing to do with the calculations which were done for  the described biogas plant. 

15.     Typically, the biogas plant is a very high cost investment and the price for the electricity from biogas plant needs subsidies to be profitable. How does it works in Burkina Faso? This project is supported by the Foundation. Would it be possible (economically feasible) to build such biogas plant as a commercial project? Please refer to this issue in the manuscript.

16.     How organic waste is obtained? Is it bought from farmer or industry? What are the expenditures for this substrate?

17.     I miss the treatment of solid digestate. If (as it is written in further part of manuscript) there is a social unacceptance of this type of fertilizer who buys it? Is it used on plantations and farms which sell? give away? the organic waste?

18.     I miss the comparison of the results to other biogas plants/projects which use human feces.

19.     Please do not use the term “human manure” Manure is an organic fertilizer. Human feces are human excrement is much better. Please check the whole manuscript.

20.     There should be a space between the word and the bracket with number of reference. Please change this in whole manuscript.

21.     There should be a space between the number and the unit, please unify this in the whole manuscript.

I understand that there are a lot of benefits from biogas plants but there are also challenges such as ensuring constant supplies of substrates, expenditures, internal GHG emissions, utilization of digestate and heat. The manuscript should be a little more critical at this point.

I recommend major revision.

Round 2

Reviewer 1 Report

Highlight changes in yellow in a next revision, please. No track changes.

Dear authors, as mentioned above, no threat changes can be visible. Otherwise it is not possible to do a proper review.

This seems obvious….

This does not solve the similarity issues.

Point 2. Now let’s go back to the main document. What happens is that similarity is also present in several places. The document being used is the same and references differ or are missing, in many cases.

Response 2: Thank you for your suggestion. Additional references have been cited for similarities in the paper

I still see no source.

Point 7. The quality of the figure is low, please assure that every figure being presented is original because of the way authors present this figure. I’m not sure it is original or where the data came from. It needs to be present at the end of the caption.

“Figure 1. Global distribution of biogas generation capacity in 2020.”

Response 7: Thank you for your suggestion. In line 122-123, the “Figure 1.” has been revised.

Again, no proper review can be done with track changes active..

Authors need to clarify in the responses without any copy paste… what was, in fact, done.

Answers are extremely unclear…

Once again answers references do not solve the similarity issues. I am also waiting for the clean version. in the next review, through the system.

I see no references being mentioned when the similarity is present:

Point 13. No reference is being mentioned immediately before the equation is presented. However, this is not new.

“causing negative impacts on global warming and climate change. Thus, the biogas project plays an important role in the GHG emission process. The baseline emission of human waste in an open lagoon was approximately 4541.63 tCO2eq per year, calculated as fol- lows:”

Response 13: Thank you for your suggestion. In line 234-242, references have been added to the manuscript.

Reference needs to be introduced immediately before the equation is presented.

Not like this.

“(28 from the IPCC [46]).”

How?!

Unclear answers:

Point 17. Authors are using previous study data to build this one. Then it needs to be highlighted, justified, explained and clarified.

Response 17: Thank you for your suggestion. We have revised

I keep my comment, it makes no sense so many headings and tiny content in each one.

Point 21. It is my perspective that the results and discussion section are just to separated a lot of Headings, tiny content in each one.

Response 21: Thank you for your professional suggestion. In this paper, we briefly introduce the opportunities and challenges affecting the development of biogas engineering in Africa, mainly from a holistic perspective, and therefore do not develop each point of study

No answers here:

Point 22. Despite finding the article important specifically in the contest of developing country, please assure that the above comemnts can be used to improve it. Tables are not that useful if the content is not better presented. Consider including some diagrams to refer to the parameters relation, for example, or to the challenges and opportunities being presented

Response 22: Thank you for your professional suggestion. In this paper, we briefly present the opportunities and challenges affecting the development of biogas engineering in Africa, mainly from a holistic perspective, and therefore do not go into great detail about the relationship between these opportunities and challenges, to which we have added some references. However, we will consider strengthening this aspect in the next studies.

I hope the authors are able to deliver a clean version with every change highlighted in a different color. I suggest yellow.

I do feel the manuscript has been improved but similarity needs to be dropped and specific suggestions attended, making the manuscript more relevant.

I do need to see lower similarity in both the manuscript as the supplementary material, in particular.

Please keep as supplementary only what needs to be, in fact, considered supplementary, otherwise consider moving  part of it to the main text and leaving as supplementary only the essential (no similarity or very little, justified).

Thus, the results and discussion need to be improved and the supplementary material moved, in part to the main text.

Also, I strongly advise the authors to highlight the novelty innovation and originality of this study.

needs to be further worked worked, in terms of scientific relevance

Reviewer 2 Report

I would like to express my sincere thanks for the effort Authors put into correcting the manuscript after the review. Authors’ work and commitment are commendable, and the corrections made in response to the reviewer' comments were very accurate and needed. Thanks to Authors’ involvement, the article has become much clearer, and its content has been supplemented with important information.

Author Response

We sincerely appreciate your feedback, which we use to improve the quality of our manuscripts. Finally, we thank you for your approval of the revised manuscript.